# Physicochemical and Functional Changes in Lotus Root Polysaccharide Associated with Noncovalent Binding of Polyphenols

**DOI:** 10.3390/foods12051049

**Published:** 2023-03-01

**Authors:** Qiulan Liu, Xiaoqin Zou, Yang Yi, Ying Sun, Hongxun Wang, Xueyu Jiang, Kaidi Peng

**Affiliations:** 1College of Food Science and Engineering, Wuhan Polytechnic University, Wuhan 430023, China; 2Hubei Key Laboratory for Processing and Transformation of Agricultural Products, Wuhan Polytechnic University, Wuhan 430023, China; 3College of Life Science and Technology, Wuhan Polytechnic University, Wuhan 430023, China

**Keywords:** lotus root, polysaccharide, polyphenol, noncovalent interaction, bioactivity

## Abstract

To promote the functional applications of lotus root polysaccharides (LRPs), the effects of noncovalent polyphenol binding on their physicochemical properties, as well as antioxidant and immunomodulatory activities, were investigated. Ferulic acid (FA) and chlorogenic acid (CHA) were spontaneously bound to the LRP to prepare the complexes LRP-FA_1_, LRP-FA_2_, LRP-FA_3_, LRP-CHA_1_, LRP-CHA_2_ and LRP-CHA_3_, and their mass ratios of polyphenol to LRP were, respectively, 121.57, 61.18, 34.79, 2359.58, 1276.71 and 545.08 mg/g. Using the physical mixture of the LRP and polyphenols as a control, the noncovalent interaction between them in the complexes was confirmed by ultraviolet and Fourier-transform infrared spectroscopy. The interaction increased their average molecular weights by 1.11~2.27 times compared to the LRP. The polyphenols enhanced the antioxidant capacity and macrophage-stimulating activity of the LRP depending on their binding amount. Particularly, the DPPH radical scavenging activity and FRAP antioxidant ability were positively related to the FA binding amount but negatively related to the CHA binding amount. The NO production of the macrophages stimulated by the LRP was inhibited by the co-incubation with free polyphenols; however, the inhibition was eliminated by the noncovalent binding. The complexes could stimulate the NO production and tumor necrosis factor-α secretion more effectively than the LRP. The noncovalent binding of polyphenols may be an innovative strategy for the structural and functional modification of natural polysaccharides.

## 1. Introduction

A high intake of plant-based natural foods is associated with the prevention of various degenerative and chronic diseases, such as obesity, diabetes, coronary heart disease, etc., and polysaccharide and polyphenol are two important components contributing to the health benefits [1]. As biological macromolecules, polysaccharides play multiple roles in the life process. Those from natural products have attracted more and more attention due to their wide variety of pharmacological activities, non-toxicity and high stability. In particular, as an immunomodulator, polysaccharides can be recognized by various immune cell receptors to trigger the different signaling pathways of the immunological response [2]. Both researchers and consumers have shown increasing interest in polysaccharides and their products for disease alleviation and health benefits [3]. Meanwhile, polyphenols have a variety of physiological functions as natural antioxidants, such as ferulic acid (FA) and chlorogenic acid (CHA). They are widely applied in food, medicine and cosmetic fields due to their excellent performances in antioxidant, hypoglycemic, anti-inflammatory and other activities [4,5].

Polysaccharides and polyphenols generally coexist in vegetables, fruits, legumes, grains, tea and other plant foods. The tissue destruction of plant foods by processing may result in the mixture of polysaccharides and polyphenols, followed by the spontaneous noncovalent interaction between them. The formation of polysaccharide–polyphenol complexes can sequentially affect the physicochemical and nutritional properties of processed foods [6]. For example, the noncovalent binding of phenolic acids to a polysaccharide reduced their bioavailability in the small intestine, and thus they could be transported to the large intestine and suffer from fermentation and metabolism by intestinal bacteria [7]. This binding was also reported to significantly affect the processing, storage, texture and edible quality of foods [8,9,10]. Therefore, understanding the occurrence and influence of the noncovalent interaction between a polysaccharide and polyphenol will help to improve various qualities of processed plant foods and is of great significance for guiding food processing.

As an important aquatic vegetable in China, lotus root possesses the cultivation area of 4.0 × 10^5^ hm^2^ and the annual yield of 1.2 × 10^7^ t [11]. It contains a variety of bioactive compounds such as polysaccharides and polyphenols, and their antioxidant, anti-inflammatory, immunomodulatory activities have attracted considerable research interest [12,13,14]. A previous study confirmed the noncovalent binding of the lotus root polysaccharide (LRP) to various polyphenols, which generate complexes with different contents of a polyphenol, as well as significant differences in the physicochemical and functional properties [15]. The noncovalent interaction principally induced by hydrogen bonding and the hydrophobic effect is closely related to the conditions of the temperature, pH, ionic strength and substrate concentration [6,16]. It allows for the accurate preparation of complexes with a desired mass ratio of polysaccharide to polyphenol, to investigate the effect of different binding degrees on the structural and functional properties of the complexes. 

In this work, CHA and FA were used to prepare the LRP–polyphenols complexes by using the equilibrium dialysis method, which had certain mass ratios of the LRP to the polyphenol. Their structural features were analyzed by spectroscopic and chromatographic methods. Meanwhile, their antioxidant ability and macrophage-stimulating activity were evaluated in vitro. The physicochemical and functional effects of polyphenol binding on the LRP were further explored.

## 2. Materials and Methods

### 2.1. Materials and Chemicals

Fresh lotus roots (cultivar Elian No. 5), which were purchased from Wuhan Jinshui Qiliang Agricultural and Sideline Products Co., Ltd. (Wuhan, China), were manually washed and sliced. The slices (thickness 5 mm) were dried by 65 °C hot air to the moisture content of about 12%, followed by grinding with a multi-functional grinder (Suzhou Jiangdong Precision Instrument CO., Ltd., Suzhou, China). The powder samples were then stored in a desiccator at room temperature.

FA (≥95%) and CHA (≥95%) were purchased from Aladdin Biochemical Technology Co., Ltd. (Shanghai, China). Ferric reducing antioxidant power (FRAP) assay kit was purchased from Biyuntian Biotechnology Co., Ltd. (Shanghai, China). Folin–Ciocalteu reagent and other analytical reagents were purchased from Sinopharm Chemical Reagent Co., Ltd. (Shanghai, China). Lipopolysaccharide (LPS) was gained from Beijing Boaotuo Technology Co., Ltd. (Beijing, China). Dulbecco’s modified eagle medium (DMEM) and phosphate-buffered saline (PBS, pH 7.4) were obtained from HyClone (Logan, UT, USA). A cell counting kit (CCK-8) was acquired from Langeco Technology Co., Ltd. (Beijing, China). The mouse TNF-α ELISA kit was supplied from Dakewe Bio-engineering Co., Ltd. (Shenzhen, China).

### 2.2. Extraction of LRP

LRP was extracted from lotus root powder using a previous method [15], with slight modifications. Briefly, the powder (100 g) was dispersed in 1000 mL distilled water at room temperature, accompanied with 400 W ultrasound treatment for 10 min. The mixture was then transferred to a 90 °C water bath. After 2 h hot-water extraction, the mixture was water cooled and added in absolute ethanol (final volume concentration 30%) to precipitate starches at 4 °C for 3 h. The supernatant was isolated by centrifugation (4500 r/min, 10 min) and added in absolute ethanol (final volume concentration 75%) to sequentially precipitate polysaccharides at 4 °C overnight. Finally, the precipitate obtained by centrifugation (4500 r/min, 10 min) was redissolved in 30 mL distilled water, and then vacuum freeze-dried to collect powdery LRP.

### 2.3. Composition Analysis of LRP

The contents of polysaccharide and protein in LRP were determined by the phenol-sulfuric acid method [17] and Coomassie blue assay kit, respectively. In addition, the monosaccharide composition of LRP was analyzed by a reversed-phase high-performance liquid chromatography method established by Yi et al. [14].

### 2.4. Preparation of LRP–Polyphenol Complexes

LRP–polyphenol complexes were prepared by an equilibrium dialysis method [15], with slight modifications. The aqueous solutions of LRP (2 mg/mL) and polyphenol (1 mg/mL) were mixed at certain conditions according to Table 1. After stirring at 120 r/min for 30 min, the mixture was transferred to a dialysis bag (MWCO 500 Da) and dialyzed against distilled water for 72 h to remove free polyphenols. Macromolecules remaining in the bag were vacuum freeze-dried and named as complex LRP-FA or LRP-CHA. In addition, LRP and polyphenols were directly mixed to obtain their physical mixtures, which were named as LRP&FA (61.18 mg FA/g LRP) and LRP&CHA (1276.71 mg CHA/g LRP).

### 2.5. Spectral Analyses

Powdery sample was dissolved in distilled water and diluted to the concentration of 50 μg/mL. The sample solution was then scanned in the range of 190~400 nm using an ultraviolet (UV) and visible spectrophotometer (TU-1810, Puxi, Shanghai, China), with the baseline established by distilled water.

Sample (1 mg) mixed with dried KBr (200 mg) was ground and pressed into tables for Fourier-transform infrared spectroscopy (FTIR) analysis. The analysis was performed with an FTIR spectrophotometer (Nexus 5DXC FT-IR, Thermo Nicolet, Madison, WI, USA), using the frequency range of 4000~400 cm^−1^, the scanning number of 32 and the resolution of 4 cm^−1^ [18].

### 2.6. Measurement of Molecular Weight Distribution

The high-performance size-exclusion chromatography coupled with multi-angle laser light scattering and refractive index detection (HPSEC-MALLS-RI) was used to analyze the molecular weight (Mw) distribution of samples according to the method of Yi et al. [19].

### 2.7. Evaluation of Antioxidant Capacity

The 1,1-diphenyl-2-picrylhydrazyl (DPPH) radical scavenging activity of samples was assessed using the method of Han et al. [20], with slight modifications. The aqueous solution of sample (1 mL, 0.1~0.8 mg/mL) was mixed with the alcohol solution of DPPH (4.0 mL, 0.2 mmol/L), followed by reaction in the dark for 30 min at room temperature. The mixture of 1 mL distilled water and 4.0 mL DPPH solution was used as the control. The absorbance of reaction mixture was finally measured at 517 nm. The DPPH radical scavenging rate (S, %) of sample solution was calculated as its decrease percentage in absorbance relative to the control. Meanwhile, the FRAP capacity of samples was evaluated using a FRAP assay kit according to its instruction, expressed as millimoles of Fe^2+^ equivalents per 1 g of sample.

### 2.8. Evaluation of Macrophage-Stimulating Activity

The effect of samples on the cell viability of RAW264.7 macrophages was evaluated using a CCK-8 kit according to its instruction. In brief, cells were harvested in the logarithmic phase of growth and dispersed in complete culture medium (DMEM containing 10% fetal bovine serum) to a cell density of 2 × 10^5^ cells/mL. Then, 200 μL/well of cell suspension was plated in 96-well culture plates for 24 h incubation at 37 °C in a humidified 5% CO_2_ incubator (MCO-17 AIC, SANYO, Tokyo, Japan). The adherent cells were washed twice with PBS after removal of culture supernatant, followed by 24 h incubation in 200 μL medium containing samples (50, 100, 200 or 400 μg/mL, four replicates for each concentration). In addition, the cells incubated without sample stimulation were used as control, and the wells containing 200 μL culture medium only were set as blank. After 2 h reaction with CCK-8 solution (20 μL/well), the optical density (OD) of each well was measured at 450 nm on a microplate reader (Westchemy Technology Co., Ltd., Beijing, China). The cell viability (%) was calculated as follows: (OD_s_ − OD_b_)/(OD_c_ − OD_b_) × 100%, where OD_s_, OD_c_, OD_b_ are the OD values of sample, control and blank, respectively.

Moreover, the effects of samples on the nitrogen oxide (NO) production and tumor necrosis factor-alpha (TNF-α) secretion of macrophages were evaluated using the method of Yi et al. [14]. Their test concentrations were 50, 100 and 200 μg/mL, while 500 ng/mL LPS was used as a positive control. After incubation with the stimulants for 24 or 48 h, the NO production in culture supernatant was analyzed by the Griess method and expressed as sodium nitrite equivalents (μmol/L) [14]. Meanwhile, the TNF-α concentration (pg/mL) was determined using a mouse TNF-α ELISA kit according to its instruction.

### 2.9. Statistical Analysis

Experimental data were expressed as “mean ± standard deviation”. Duncan method was adopted to analyze the significant differences between groups at the 0.05 level by SPSS19.0 software (Chicago, IL, USA).

## 3. Results and Discussion

### 3.1. Basic Composition of LRP

It has been reported that polysaccharides from various cultivars and parts of lotus root differed significantly in content, composition and structure [21]. Of the 13 cultivars, Elian No. 5, which showed a relatively high polysaccharide content, was selected to prepare the LRP. The polysaccharide content and protein content of the LRP were 83.27% and 1.46%, respectively. The polysaccharide content was obviously higher than the previous result of 71.31% [21], which might be related to the preparation difference for the polysaccharides. It was suggested that the method of two-step alcohol precipitation in the present work was more effective for isolating the LRP with a higher purity, as compared to the traditional combination of amylase-based enzymolysis, the Sevage method and 75% alcohol precipitation. The LRP was composed of rhamnose, glucose, galactose and arabinose at the molar ratio of 0.12:7.63:1.30:0.95. This composition was approximately in accord with the previous results [17,21]. 

### 3.2. Factors Influencing the Formation of LRP–Polyphenol Complexes

LRP–polyphenol complexes were designed by controlling the conditions of the temperature, pH and polysaccharide-polyphenol mass ratio for a noncovalent interaction. The polyphenol binding amount (mg/g LRP) in the serial complexes of both the LRP-FA and LRP-CHA showed an approximate ratio of 4:2:1 (Table 1). Compared to the FA, the CHA showed a stronger affinity to the LRP, which might be closely related to its galloyl group [22]. Tang et al. [22] indicated that the galloyl group was the major functional group in the cellulose–tannin interactions. Its three hydroxyl groups potentially contributed to the hydrogen bonding, and the aromatic moiety of the tannin was important for the hydrophobic interactions. At the same time, because the formation of a hydrogen bond is an exothermic process, the decrease in the binding amount caused by a temperature rise might be due to the weakening of the hydrogen bond [23,24]. In addition, when the temperature increased, the hydrophobic interaction between the benzene rings was conducive to the formation of FA polymers [25], which may hinder the noncovalent binding between the FA and LRP. The mixing ratio of polysaccharides and polyphenols is also an important factor affecting their interaction [26]. With the increase in the mass ratio of CHA to LRP in the mixed system, the availability of the active sites of LRP molecules was conducive to increasing the binding amount [27]. Moreover, the change in the pH would lead to the change in the potential on the molecular surface of polysaccharides and polyphenols, affecting the electrostatic interaction between them, and thus also affecting the noncovalent binding of them [28].

### 3.3. Spectral Features of LRP–Polyphenol Complexes

The characteristic UV peaks of polyphenols mainly exist in the wavelength range of 240~380 nm, while most polysaccharides have no absorption in this range [29]. Therefore, it is possible to confirm the successful binding by comparing the UV spectra of polyphenols, polysaccharides and their corresponding complexes. FA was successfully adsorbed by arabinan-rich pectic polysaccharide in Zhang’s study, which was determined via UV spectra [25]. The formation of noncovalent complexes between them can result in the weakness or disappearance of the UV characteristics belonging to the polyphenols [30]. As shown in Figure 1, the LRP had a strong absorption only in the range of 190~220 nm, which was related to its unsaturated carbonyl and carboxyl groups [25]. The characteristic peaks of the FA and CHA at 312, 288 and 218 nm could be assigned to the π-π* transition of the double bond, phenol group and phenyl ring, respectively [19,25]. They were found in the spectra of the LRP&FA and LRP&CHA mixtures but were obviously weakened (even disappeared) in the spectra of the LRP-FA and LRP-CHA complexes. Thus, the results implied that the interaction of the LRP and polyphenols exists, which obviously altered the ultraviolet properties of the FA and CHA. The results confirmed the intermolecular interaction between the LRP and polyphenols in the complexes.

FTIR spectroscopy can provide structural information related to polysaccharide–phenol interactions [16]. As seen in Figure 2, the typical polysaccharide peaks of the LRP were confirmed [31], which included the stretching vibration (SV) of O–H at 3382 cm^−1^, the SV of C–H at 2931 cm^−1^, the SV of C=O at 1618 cm^−1^ and the SV of C–O at 1410 cm^−1^. Among them, the broad peak at 2931 cm^−1^ was attributed to the C–H stretching vibration of the LRP, which was also observed in the FTIR spectrum of the LRP–polyphenol complexes and mixtures. In addition, the SV of O–H in the LRP–polyphenol complex spectrum (at 3417 cm^−1^ and 3416 cm^−1^ for LRP-FA_2_ and LRP-CHA_2_, respectively) were broader than that in the LRP spectrum. It has been reported that the O–H stretching band was a sensitive indicator of the strength of the hydrogen bond, the O–H stretching band also shows a drastic change in its wave number and intensity depending on the formation of the hydrogen bonds and the broadening of the O–H stretching band was often observed in strong hydrogen bonds [32]. Here, it was suggested that the hydrogen bond was formed between the LRP and FA/CHA. 

In addition, the LRP&FA and LRP&CHA showed the characteristic bands of the aromatic nucleus in the wavenumber range of 1630~1516 cm^−1^, which were assigned to the SV of the aromatic ring C=C [33]. The LRP&FA and LRP&CHA mixtures spectrum had characteristic peaks at 1604 cm^−1^ and 1522 cm^−1^ and at 1633 cm^−1^, 1597 cm^−1^ and 1516 cm^−1^, respectively, which were weakened or even disappeared in the corresponding complex. Moreover, the bands at 1275 cm^−1^ and 1031 cm^−1^ were both related to the C−O−C vibration, and the bands near 1123 cm^−1^ and 618 (or 612) cm^−1^, respectively, belonged to the bending stretching (BS) of the phenolic hydroxyls [34] and the C–H bonds of the out-of-plane flexion in the aromatic compounds [19]. By comparing with the mixtures, the LRP–polyphenol complexes showed significantly attenuated bands at 1123 cm^−1^ and 617 cm^−1^. Moreover, the absorption intensity of the bands at 852 cm^−1^, 811 cm^−1^, 849 cm^−1^ and 816 cm^−1^ was decreased. The attenuated signals of the phenolic hydroxyls and rings indicated the noncovalent interaction between the LRP and polyphenols, probably via the hydrogen bonding and hydrophobic interaction. 

### 3.4. Molecular Weight Distribution of LRP–Polyphenol Complexes

For the LRP, the increase in the Mw is one of the typical features associated with the noncovalent binding of polyphenols [15]. The effect of the polyphenol-binding ratio on the Mw distribution of the LRP–polyphenol complexes was determined by HPSEC-MALLS-RI. The chromatograms are shown in Appendix A, and the detailed information is collected in Table 2. The LRP consisted of seven fractions, with an average Mw of 146.3 kDa, of which the faction with the highest Mw of 1144 kDa accounted for 8.4% of the total peak area; by comparison, the LRP-FA and LRP-CHA complexes showed obviously higher values of the average Mw. Of all the fractions, the one with the highest Mw (1627 kDa, 14.1%) was found in LRP-FA_3_. The LRP contained 52.5% fractions within the low Mw range of 3.0 ≤ lg Mw < 4.0 (Figure 3). The low Mw fractions in the LRP-CHA_1_, LRP-CHA_2_ and LRP-CHA_3_, respectively, accounted for 4.7%, 2.8% and 5.6%, while those in the LRP-FA complexes were not detected. Moreover, the relative content of the fractions within the range of 4.0 ≤ lg Mw < 4.5 and 5.0 ≤ lg Mw < 5.5 significantly increased in all the complexes. No fraction below 1 kDa was found in the complexes, indicating that they did not contain free FA and CHA.

Compared to the CHA, the FA with a lower affinity for the LRP could form complexes with higher Mws. Moreover, the increase in the polyphenol binding reduced the Mw of both the LRP-CHA and LRP-FA complexes. It was suggested that more polyphenols combined to the single molecule of the LRP might form the cluster structure with a relatively strong effect of the steric hindrance, inhibiting the unutilized active groups of polyphenols noncovalently bound to other polysaccharide molecules. 

### 3.5. Antioxidant Activity of LRP–Polyphenol Complexes

Previous studies indicated that the antioxidant activity of the LRP was relatively weak. Its IC_50_ value of DPPH radical scavenging was 1.65 mg/mL [17]. As natural antioxidants, phenolic compounds still retained their antioxidant activity when combined with soluble dietary fiber [35], and the antioxidant activity of their complex might be positively correlated with the phenolic binding amount [15]. As shown in Figure 4A,B, the noncovalent binding of polyphenols could effectively enhance the DPPH radical scavenging activity of the LRP. The scavenging capacity of the LRP-FA was significantly higher than that of the LRP (*p* < 0.05) and was positive to the FA binding amount. It might be attributed to the increased number of active hydroxyl groups, providing more protons to block the free-radical reaction [36]. Alejandra et al. also obtained similar results that a phenolic antioxidant (3,4-Dihydroxyphenylglycol, DHPG) linked to soluble dietary fiber maintains some antioxidant activity, and it was explained that the additional -OH group of the DHPG provided the availability of the catechol group and contributed to the antioxidant activity of the soluble fiber–DHPG complex. By comparison, the LRP-CHA complexes with higher polyphenol proportions did not show a prominent advantage in the DPPH radical scavenging. Moreover, their scavenging ability decreased with the increase in the CHA binding amount. The results implied that the active hydroxyl groups of the CHA in the complexes were significantly consumed due to the formation of intermolecular hydrogen bonds or inhibited by a steric hindrance. In addition, the LRP–polyphenol complexes showed a lower scavenging capacity than their physical mixtures, due to the reduction in active hydroxyl groups by the hydrogen bond formation [37]. The DPPH radical scavenging IC_50_ values of the LRP, LRP-FA_1_, LRP-CHA_3_, LRP&FA and LRP&CHA were 1.04, 0.47, 0.58, 0.17 and 0.10 mg/mL, respectively. 

The FRAP capacities of the LRP and LRP–polyphenol complexes were also evaluated, as seen in Figure 4C. The FRAP value of the LRP was 0.29 mmol/g, which showed no significant difference with the LRP-FA complexes (*p* > 0.05). In comparison to them, the LRP-CHA_2_ and LRP-CHA_3_ possessed stronger FRAP capacities (*p* < 0.05), and the capacity showed a negative correlation with the CHA binding amount which was similar to the DPPH radical scavenging activity. In addition, the physical mixtures of the LRP and polyphenols exhibited higher FRAP values than their corresponding complexes (*p* < 0.05), indicating that the noncovalent interaction significantly decreased their FRAP capacity.

### 3.6. Macrophage-Stimulating Activity of LRP–Polyphenol Complexes

Macrophages have a unique role in the immune system because they not only elicit an innate immune response but are also effector cells in inflammation and infection [38]. After incubation with the LRP or LRP–polyphenol complexes (50~400 μg/mL) for 24 h, the proliferation rates of the macrophages indicated that they had no cytotoxicity. Their immunostimulatory effects on the cells were evaluated in the concentration ranges of 50~200 μg/mL.

Macrophages kill pathogens not only directly through phagocytosis but also indirectly through the secretion of pro-inflammatory cytokines. When macrophages are activated, they release NO to kill microorganisms, parasites and tumor cells while inducing an inflammatory response to protect the body [39]. As seen in Figure 5A,B, the LRP and LRP–polyphenol complexes all stimulated NO production as compared to the control (*p* < 0.05), and the 48 h incubation had almost double the NO production of the 24 incubation. The FA and CHA had relatively weak effects on the NO production, and they both showed inhibitory effects on the immunostimulation of the LRP. However, the inhibition disappeared after the noncovalent binding of polyphenols to the LRP. 

TNF-α is an immunoactive factor responsible for regulating the response of inflammation and the host defense, and it plays a role in the removal of pathogens and the promotion of wound healing [40]. As seen in Figure 5C, the LRP could also stimulate TNF-α secretion as compared to the control (*p* < 0.05), but the effect was significantly weaker than those of the LRP–polyphenol complexes (*p* < 0.05). Overall, the LRP-FA_1_ and LRP-CHA_3_ exhibited relatively strong immunostimulation on the macrophages. The difference may be attributable to discrepancies in the active compounds (polyphenol) [41].

## 4. Conclusions

LRPs, as the main bioactive macromolecules in lotus root, can spontaneously combine with FA/CHA through a noncovalent interaction. The resulting complexes showed significant differences in the structural and functional features as compared to the LRP. The average Mw of the LRP–polyphenol complexes was 1.11~2.27 times higher than that of the LRP. The LRP-FA/CHA complexes and their mixtures had a significantly stronger DPPH radical scavenging activity and FRAP capacity than the LRP, which was due to the increased number of active hydroxyl groups of the FA/CHA, and the effects were closely related to the structure and binding amount of the polyphenols. However, the LRP–polyphenol complexes showed lower antioxidant activity than their mixtures, due to the reduction in active hydroxyl groups by the hydrogen bond formation. Moreover, the binding had positive effects on the macrophage activation, by stimulating the production of NO and TNF-α. Free polyphenols could inhibit the LRP-induced NO production, but this inhibitory effect disappeared after the formation of the noncovalent binding. This work provides a promising approach to improve the functional activity of polysaccharides and the bioavailability of polyphenols by modulating the noncovalent interactions between them. However, endogenous and exogenous factors affecting the bioactivity and bioavailability of polysaccharide–polyphenol complexes, as well as the structure–activity relationship, need to be additionally explored.

## Figures and Tables

**Figure 1 foods-12-01049-f001:**
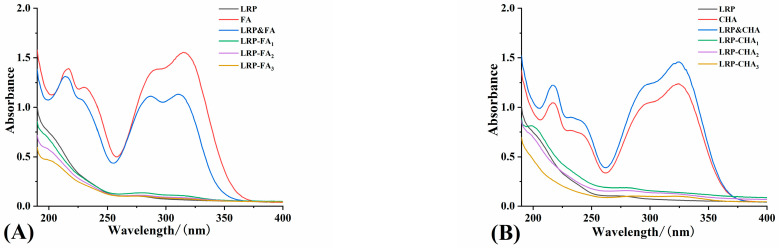
UV spectra of LRP-FA complexes (**A**) and LRP-CHA complexes (**B**).

**Figure 2 foods-12-01049-f002:**
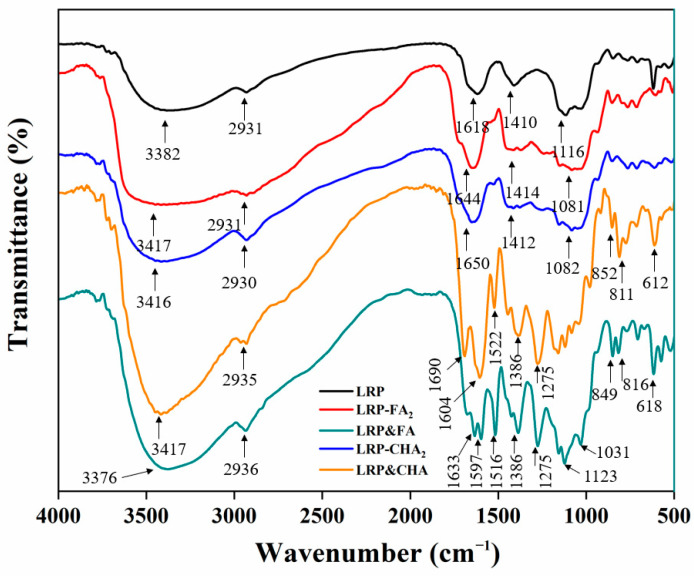
FTIR spectra of LRP and LRP–polyphenol complexes.

**Figure 3 foods-12-01049-f003:**
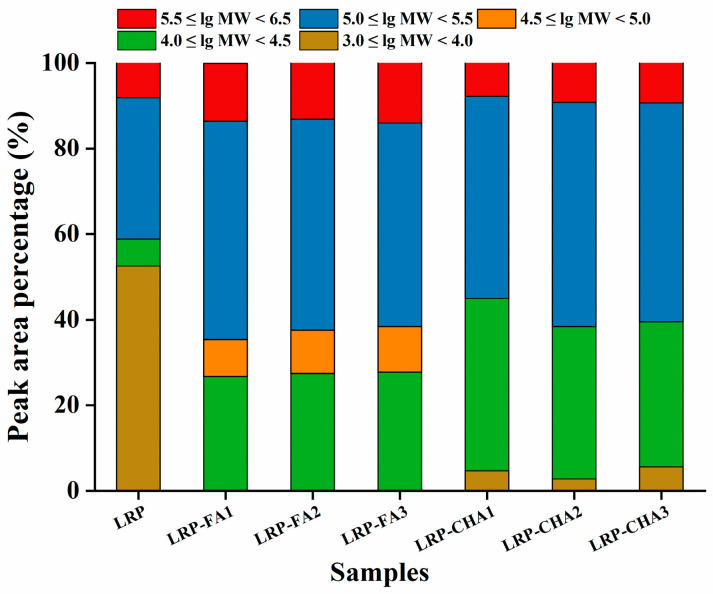
Comparison of the molecular weight distribution of LRP and LRP–polyphenol complexes.

**Figure 4 foods-12-01049-f004:**
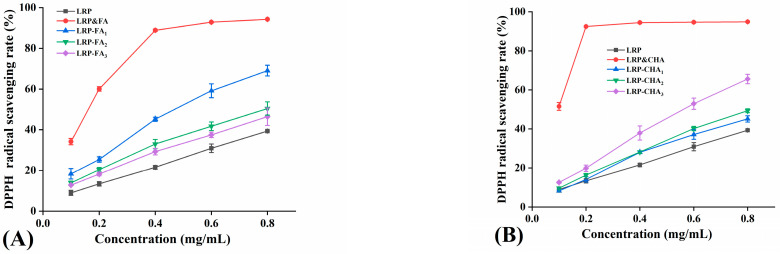
Antioxidant activity of LRP and LRP–polyphenol complexes. (**A**,**B**) are the DPPH radical scavenging activity, and (**C**) is the FRAP total antioxidant capacity. (a–e indicates that there are significant differences between different letters, *p* < 0.05).

**Figure 5 foods-12-01049-f005:**
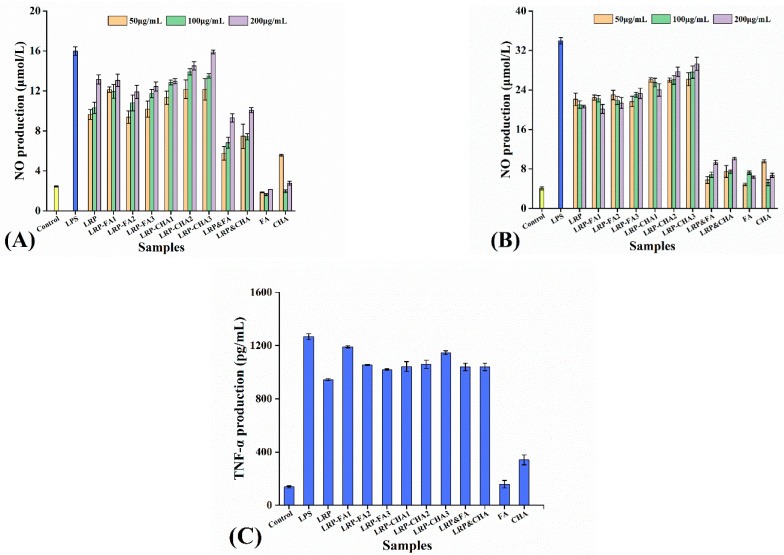
Effect of LRP and LRP–polyphenol complexes on macrophage activation. (**A**,**B**) are the NO production after 24 h and 48 incubations, respectively. (**C**) is the TNF-α secretion after 24 h incubation.

**Table 1 foods-12-01049-t001:** Preparation conditions of LRP–polyphenol complexes and their polyphenol binding amount.

Complexes	Preparation Conditions			Polyphenol Binding Amount (mg/g LRP)
	Temperature (°C)	pH	Mass Ratio of LRP to Polyphenol	
LRP-FA_1_	0	6	2:1	121.57
LRP-FA_2_	15	5	2:1	61.18
LRP-FA_3_	30	5	2:1	34.79
LRP-CHA_1_	15	5	1:4	2359.58
LRP-CHA_2_	15	5	1:2	1276.71
LRP-CHA_3_	15	6	1:1	545.08

**Table 2 foods-12-01049-t002:** Molecular weight distribution of LRP and LRP–polyphenol complexes.

Samples	Retention Time (min)	Molecular Weight (Da)	Peak Area Percentage (%)	Average Molecular Weight (Da)
LRP	7.297–9.107	1.144 × 10^6^ (±0.791%)	8.4	1.463 × 10^5^
	9.077–11.490	1.359 × 10^5^ (±1.050%)	33.1	
	11.490–12.184	2.628 × 10^4^ (±4.650%)	6.3	
	12.184–13.934	9.036 × 10^3^ (±3.661%)	20.6	
	13.934–14.718	3.695 × 10^3^ (±5.083%)	12.7	
	14.748–18.398	6.798 × 10^3^ (±4.744%)	18.3	
	21.868–24.402	2.494 × 10^3^ (±51.551%)	0.9	
LRP-FA_1_	7.026–8.926	1.465 × 10^6^ (±0.723%)	13.5	2.978 × 10^5^
	8.926–11.400	1.730 × 10^5^ (±0.858%)	50.3	
	11.400–12.637	2.898 × 10^4^ (±4.228%)	19.5	
	12.637–13.994	4.509 × 10^4^ (±4.033%)	7.5	
	13.994–15.291	3.066 × 10^4^ (±3.972%)	7.2	
	15.291–15.985	7.438 × 10^4^ (±4.292%)	1.2	
	17.373–19.002	1.283 × 10^5^ (±3.532%)	0.7	
LRP-FA_2_	7.237–8.896	1.603 × 10^6^ (±0.721%)	13.1	3.102 × 10^5^
	8.896–11.340	1.820 × 10^5^ (±0.824%)	48.7	
	11.340–12.516	2.681 × 10^4^ (±3.786%)	19.3	
	12.516–13.873	3.354 × 10^4^ (±4.439%)	8.8	
	13.873–15.261	2.407 × 10^4^ (±3.966%)	8.1	
	15.261–16.076	5.990 × 10^4^ (±3.650%)	1.4	
	17.312–18.821	1.179 × 10^5^ (±4.124%)	0.6	
LRP-FA_3_	7.297–8.926	1.627 × 10^6^ (±0.688%)	13.1	3.321 × 10^5^
	8.926–11.309	1.894 × 10^5^ (±0.735%)	48.7	
	11.309–12.516	2.956 × 10^4^ (±2.849%)	19.3	
	12.516–13.904	3.674 × 10^4^(±2.926%)	8.8	
	13.904–15.261	2.723 × 10^4^ (±2.523%)	8.1	
	15.261–16.226	7.541 × 10^4^ (±2.588%)	1.4	
	17.343–18.549	2.141 × 10^5^ (±2.951%)	0.6	
LRP-CHA_1_	7.207–8.956	1.157 × 10^6^ (±0.660%)	7.8	1.621 × 10^5^
	8.956–11.370	1.365 × 10^5^ (±0.960%)	47.2	
	11.400–12.576	1.840 × 10^4^ (±5.402%)	19.8	
	12.576–13.934	2.211 × 10^4^ (±5.505%)	8.2	
	13.934–15.020	1.253 × 10^4^ (±5.063%)	7.5	
	15.020–16.317	1.676 × 10^4^ (±4.756%)	4.8	
	17.011–18.700	4.540 × 10^3^ (±23.036%)	4.7	
LRP-CHA_2_	7.177–8.931	1.184 × 10^6^ (±0.671%)	9.2	1.913 × 10^5^
	8.931–11.392	1.424 × 10^5^ (±0.955%)	52.4	
	11.392–12.561	2.015 × 10^4^ (±5.290%)	20.8	
	12.561–13.953	2.674 × 10^4^ (±4.907%)	7.8	
	13.953–15.485	1.875 × 10^4^ (±3.172%)	7.0	
	16.837–18.773	7.451 × 10^3^ (±25.715%)	2.8	
	7.177–8.931	1.184 × 10^6^ (±0.671%)	9.2	
LRP-CHA_3_	7.237–8.866	1.545 × 10^6^ (±0.691%)	9.4	2.433 × 10^5^
	8.866–11.340	1.724 × 10^5^ (±0.791%)	51.2	
	11.340–12.546	2.422 × 10^4^ (±4.184%)	20.4	
	12.546–13.964	3.613 × 10^4^ (±4.401%)	7.5	
	13.964–15.291	2.728 × 10^4^ (±3.526%)	6.0	
	16.528–19.122	8.365 × 10^3^ (±22.813%)	5.6	

## Data Availability

Data is contained within the article.

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
