# Peer review of "Physicochemical and Functional Changes in Lotus Root Polysaccharide Associated with Noncovalent Binding of Polyphenols"

_foods, 2023, doi:10.3390/foods12051049_

Round 1

Reviewer 1 Report

Dear authors, please read the file attached.

Author Response

Dear reviewer,

Many thanks for your fast reply and your valuable comments on scientific and linguistic aspects, which are of great help to the improvement of the article.  Thank you very much!

The following document is what I replied to you, please check it. 

Reviewer 2 Report

Title: Physicochemical and functional changes of lotus root polysaccharide associated with non-covalent binding of polyphenols

Authors: Yang Yi and co-workers

Manuscript ID: foods-2198484

In this manuscript, LRP-polyphenol complexes were prepared by equilibrium dialysis method and evaluated the physicochemical and bio-functional properties. This research work is very interesting and the manuscript also well prepared. However, before publication, authors should address the following clarifications, I recommend major revision.

Major revision

1.      Line 107; The contents of polysaccharide and protein in LRP were determined by the phenol-107 sulfuric acid method… : Provide the determined data in the supporting information for the ready reference

2.      LRP-FA and LRP-CHA UV-vis studies, authors did not account the blue shift observed in LRP-FA. If both the complexes are soluble in the solvent taken, then it should show absorbance with some shift. Why there is no absorbance for polyphenol in complex?

3.      Authors should include the molecular structures for clarity

4.      FTIR characteristic region expansion should be provided and the shift should be denoted in the expanded spectra.

5.      Authors should justify how the absorbance and stretching frequency of polyphenol got disappeared in UV-vis and FTIR. Line 232: “were also weakened or even disappeared” is not a concrete result, because the complex characterization is completely relay on the above two methods

6.      How the non-available polyphenol in complex displayed antioxidant property

7.      Pictorial representation for the formation of complex and the activity should be provided

8.      Spectra used to frame table 2 should be provided in ESI

9.      Picture of antioxidant assay should be provided in ESI

10.  Minor revision: Line 121: No space between (° C)

 Author Response

Dear reviewer,

Thank you for your positive comments and valuable suggestions to improve the quality of our manuscript.

The following document is what I replied to you, please check it. 

Round 2

Reviewer 2 Report

I am satisfied with the revision made and hence I recommend publication of this manuscript in “Foods”.